# School-Based Interventions to Support Healthy Indoor and Outdoor Environments for Children: A Systematic Review

**DOI:** 10.3390/ijerph20031746

**Published:** 2023-01-18

**Authors:** Amanda Fernandes, Mònica Ubalde-López, Tiffany C. Yang, Rosemary R. C. McEachan, Rukhsana Rashid, Léa Maitre, Mark J. Nieuwenhuijsen, Martine Vrijheid

**Affiliations:** 1ISGlobal, Barcelona Institute for Global Health, 08003 Barcelona, Spain; 2Universitat Pompeu Fabra (UPF), 08003 Barcelona, Spain; 3CIBER Epidemiología y Salud Pública (CIBERESP), 28029 Madrid, Spain; 4Bradford Institute for Health Research, Bradford Teaching Hospitals NHS Foundation Trust, Bradford Royal Infirmary, Bradford BD9 6RJ, UK

**Keywords:** air pollution, road-traffic noise, urban green space, active travel, systematic review, school children

## Abstract

Environmental exposures are associated with children’s health. Schools are often urban exposure ‘hotspots’ for pollution, noise, lack of green space and un-walkable built environments. The aim of this systematic review was to explore the impact of school-based interventions on the modification of indoor and outdoor stressors related to the built and natural environment on children’s exposure and health. A systematic review of seven databases was performed. We included quantitative studies on children aged 5–12, which reported intervention delivered within school settings aimed at addressing key environmental exposures including air pollution, green spaces, traffic noise or active travel; and reported physical and mental health, physical activity or active travel behavior. The quality of studies was assessed and interventions were described using a standardized framework. A narrative synthesis approach was used to describe the findings. Thirty-nine papers were included on three main intervention types: improve indoor air quality by the increase of ventilation rates in classrooms; increase children’s green time or greening schools, and multicomponent interventions to increase active travel to school by changes in pedestrian facilities. No eligible intervention to reduce traffic noise at school was found. Increasing ventilation rates improved short-term indoor air quality in classrooms, but the effect on cognitive performance was inconsistent. Greening schools and increasing children’s green time have consistent positive effects on cognition and physical activity, but not in behavior. Multi-component interventions can increase walking and cycling after three years. Overall, the studies were rated as having poor quality owing to weak study designs. We found modest evidence that school-based built and natural environment interventions can improve children’s exposure and health.

## 1. Introduction

Children are particularly vulnerable to their urban environment, a source of physical, chemical, and behavioral exposures (e.g., air and noise pollution, lack of green spaces, and physical inactivity) all of which have been associated with a variety of cardiovascular, respiratory, and neurodevelopmental problems [1,2,3]. In Europe, children spent between 170–190 days per year at primary school, which is the most important micro-environment after home [4]. Schools are often urban exposome [5] ‘hotspots’ located in dense areas of high pollution and noise with scarce vegetation compounded by high levels of car use during the ‘school run’ [4].

School buildings are usually close to roads with insufficient insulation structures to prevent contamination from outside. Most of them rely on manual airing providing inadequate ventilation rates in classrooms, which become worse during heated seasons [6]. Excess vehicle traffic and lack of built and natural facilities may create unsafe and unfriendly environments for walking and biking, which in turn increase the number of children driven by private vehicles to school and consequently, air pollution (e.g., vehicle idling) [4]. The prevalence of physical inactivity among primary-school children is a public health burden, and about 40% do not actively commute to or from school in European cities [7]. Therefore, exposure to urban stressors in school settings has an impact on child health with consequences to development, learning, and ultimately academic performance.

Vehicle-source (e.g., NO_2_ and O_3_) and indoor-generated pollutants (e.g., CO and volatile organic compounds) induce and exacerbate acute (e.g., respiratory irritation) and chronic (e.g., asthma, allergies, and lung function decay) respiratory problems in school children [3]. High concentrations of NO_2_, PM_2.5_, elemental carbon, and ultrafine particles in schoolyards have been associated with increased odds of overweight or obesity in children aged 7–10 years [8] well as smaller growth in their working memory [9,10]. Aircraft noise at primary schools has been associated with a negative linear impact on standardized test results, and worse long-term memory, and reading comprehension in children [11]. Indeed, road traffic noise can disturb pupils’ and teachers’ interactions in the classroom leading to miscommunication, stress-mediated annoyance, inattention, and behavioral problems [12,13].

Recommendations to reduce these urban threats in the school environment include the installation of buildings far from roads, maintenance, and improvements of the filtration system, or retrofitting building heating, ventilation, and air conditioning (HVAC) design [14,15]. Particularly, the amount of surrounding vegetation might mitigate health damage through the filtration of some pollutants and buffering noise [16,17]. There is also evidence of the short and long-term restorative effects on children’s cognition functions of spending time in or even having views of parks, urban forests, or surrounding vegetation [18,19,20]. Based on those benefits, outdoor classes in nature and greening schoolyards have been raised as a strategy to foster learning and create a healthy school environment [17,21]. Furthermore, green spaces may improve physical activity through free play and enhance social interactions suggesting protective effects on behavioral problems [20].

Related to this, changes in the built environment around schools by the provision of sidewalks, green routes, and other facilities may support active travel—defined as walking or cycling as a means of transport—as a source of physical activity and reduction of air pollution [4]. Accumulative evidence from cross-sectional and longitudinal studies has associated features of the built environment with reductions in BMI [1,22,23] and blood pressure [24] in children. Notably, streetscapes supportive of walking and street aesthetics were associated with an increase in physical activity and active travel to school [23,25]. For neurodevelopment and respiratory health, most of the evidence is from adults, and the impact of built environment on children remains inconclusive [1,23,26].

A recent systematic review has summarized the impact of multi-setting air pollution interventions on morbidity and mortality, but did not include cognitive and behavioral outcomes, a key health component for children’s development [27]. Systematic and narrative reviews have mainly focused on single exposures [4], residential environments [25], streets [28], or were restricted to few health outcomes [29]. The evidence from these reviews is limited, the authors highlighted the lack of intervention-based research in this area particularly in children [25,27,28,29]. To date, there are no systematic reviews of school-based interventions combining and comparing different strategies.

Based on a broad and systematic search of current literature, this systematic review aims to summarize existing evidence from school-based intervention studies which involved changes to indoor and outdoor stressors related to the built and natural environment and reported outcomes on children’s health, health behaviors, and exposure levels. We describe the features and quality of interventions targeting children in primary schools in European and high-income countries. Specifically, we answer the following research question: what is the effectiveness of interventions with a focus on changing the built and natural school environment, both indoor and outdoor, on (1) improving physical and mental health outcomes such as body composition, asthma symptoms, and cognition, (2) health behaviors such as physical activity and active travel, and (3) levels of exposure to air pollutants, noise, and green spaces in children aged 5 to 12 years? This holistic approach may be useful to inform environmental health researchers and policymakers on how interventions can act together to improve child health.

## 2. Methods

### 2.1. Scope of the Systematic Review

We created a Population Intervention Comparison Outcomes (PICO) statement to guide the review process as follows:Population: primary school-aged children between 5 and 12 years of age enrolled in primary schools from urban areas of Europe and high-level income countries in the rest of the world.Intervention: Any intervention which involved changes to the school (indoor or outdoor) built or natural environment to reduce exposures levels to key air pollutants, road traffic or aircraft noise; to increase the visibility, availability, accessibility, or time children spent in green spaces; or to promote active travel to schoolComparator: Any intervention study was included—the comparison was expected to be no intervention, another intervention, a non-exposed control group or less exposed control groups, standard or currently existing interventions. However, we did not exclude studies based on the comparison.Outcomes: Child health outcomes (i.e., respiratory, cardiometabolic, cognitive/behavioral domains), behaviors related to physical activity and active travel, or changes in the exposure levels.

Details on the intervention and outcomes components frame in this systematic review were presented in Figure 1. We developed this scheme to structure and guide the search strategy and review process. Changes in built and natural school environments are expected to have an impact on most or all students in those places acting on more distal factors compared to interventions focused only on individual behavior changes [30].

### 2.2. Search Strategy

PubMed and EMBASE (Medical and Biomedical sciences), PsycINFO (Behavioral and Social sciences), Scopus (Physical sciences, Health Sciences, Social Sciences and Life Sciences), Green FILE, Transportation Research Information Services, and Web of Science (multidisciplinary, global warming, green building, pollution, sustainable agriculture, renewable energy, recycling, field of transportation) were used to identify peer-review original articles published in English between 1 January 2010 to 10 May 2022 to provide up-to-date evidence. The selected keywords were related to three domains: ‘population’, ‘setting’, and ‘intervention/exposure’ based on the Figure 1, previous exposure-specific reviews [1,4,25,27,28,29] and Medical Subject Headings terms. The final list was further validated by co-authors based on their knowledge of the literature in their expertise. In addition, two independent librarians validated the search terms and syntax. We first designed the search strategy in PubMed by combining the three domains using Boolean operators and then adapted it for each remaining database (Appendix A). A snowballing search on the reference lists of included articles was also conducted. More information on the study protocol can be found elsewhere (PROSPERO—CRD42020187668).

### 2.3. Study Inclusion Criteria

Studies were eligible for full-text review if they included: (i) Primary school-aged children between 5 and 12 years of age enrolled in primary schools. This age band encompasses a period that is vulnerable to both adverse exposure effects and developmental adaptations [5]. Furthermore, it corresponds to the school grade period related to children’s gains in autonomy and choices to travel to school. Furthermore, behaviors, perceptions, and needs change when children become preadolescents and move to secondary school [31]. Studies that had a mixed age range were eligible if the mean age of participants was between 5–12 years at the start of the intervention; (ii) interventions implemented within, around, or along the path to the school building, for a whole class, or the entire school regardless of who delivered the intervention; (iii) schools from urban areas located in Europe listed in geographic European regions used by the Statistics Division of the United Nations [32] and high-income countries outside Europe based on World Bank classification for the 2021 fiscal year [33]; (iv) a permanent or temporary change or intervention to the built or natural school environment; (vi) objective or self-reported measures of children’s health (i.e., cardiometabolic, respiratory, cognitive, and behavior), physical activity, or active, non-motorized, travel to school; (vii) objective or self-reported measures of changes in exposure-related outcomes to air pollution, road traffic noise, and green spaces; (viii) peer-review papers published in English from 2010 onwards (Figure 1).

We excluded studies if they reported: (i) children with a critical illness or commodities; (ii) exclusively primary schools in rural areas; (iii) exclusively behavioral or educational interventions without an environment change (i.e., walking bus, reducing vehicle idling near schools, back-street walk to school, or walk day); (iv) focused on changes to other microenvironments outside school such as school bus cabins, home, neighborhoods, or community environment for a wider public; (v) pilot or case study designs, reviews, or protocols; (vi) not published in English. Conference papers, books, and grey literature were not eligible for inclusion but were inspected for relevant references. Studies using schools as the source of recruitment, but where the intervention was not school based, were excluded.

In addition, multi-component interventions were only eligible if they included an environment component with a specific target on the exposures included in this review. For instance, we excluded interventions to increase physical activity by reshaping multiple features of school playgrounds (i.e., schoolyard markings, toys provision, and replacement of some asphalt areas with artificial or natural grass [34]), but without an explicit aim to reduce air pollution or noise, or improve the exposure to green spaces. Garden interventions focusing on health outcomes through changes in food intake (e.g., salt reduction on blood pressure levels) were excluded [35]. We also excluded effects on mood (i.e., motivation, sadness, self-esteem, and stress) as psychological outcomes, because our focus was on behavioral disorders (i.e., internalization and externalization problems) more than on their transient symptoms. Laboratory studies simulating real-world conditions were also excluded.

**Figure 1 ijerph-20-01746-f001:**
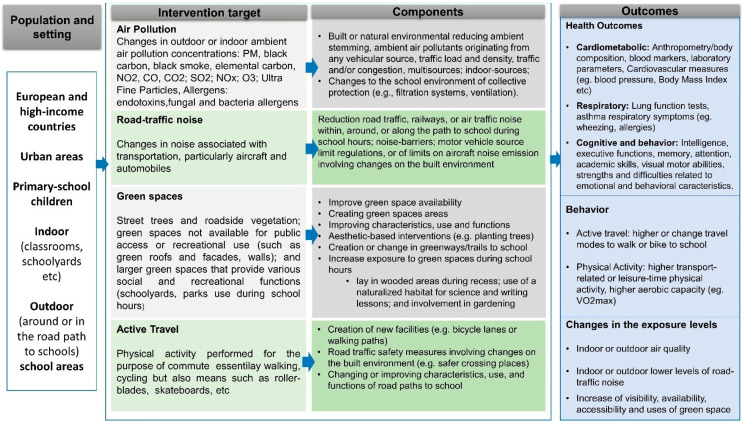
PICO-driven question scheme.

### 2.4. Data Extraction

Two independent reviewers (AF and RR) screened abstracts of studies retrieved using Rayyan QCRI^®^. Full texts of relevant articles were obtained and assessed for eligibility. We successfully contacted five authors to request additional information and to request closed-access papers. In all stages, any disagreement was resolved through discussions with a third researcher (MV). Two independent reviewers (AF and MU) conducted the data extraction in COVIDENCE^®^ and included information on the type of school, year of the study, funding sources, country, population, school characteristics, study design [35], outcome measurement, main findings, and limitations. Specifically, we extracted intervention features according to the template for intervention description and replication (TIDieR) checklist [36] to provide more detailed information on intervention characteristics and replication (see Appendix A).

### 2.5. Quality Assessment

Study quality was assessed by a tool adapted for intervention studies [37] that have previously been used in environmental reviews [28,38,39]. All included studies were assessed for quality by the first reviewer (AF) and 20% of the studies were chosen by random number generation and checked by the second reviewer (TY) with an 86% of agreement—disagreements were resolved by discussion. One point was awarded if the study met the criteria, thus studies could score between 0–11 points. A study scoring ≥ 9 was considered high-quality [28,37]. The evidence appraisal systematically assessed the risk of bias and uncertainty of the evidence. However, we did not exclude studies based on the quality score considering previous evidence on the few numbers of high-quality studies inherent to the challenges of evaluating urban interventions [28,37,39].

## 3. Results

### 3.1. Study Selection

The results of the included studies are shown in Figure 2 and followed the recommendations of the PRISMA Flow Diagram [40]. First, we retrieved 8642 papers, of which 3042 were duplicates. From a total of 5600 remaining records, 77 full texts were deemed potentially relevant. Finally, 30 studies met the eligibility criteria and were included in the review. Additionally, 9 papers were identified via the snowballing process, totalizing 39 studies: 10 targeting Air Pollution [41,42,43,44,45,46,47,48,49,50], 17 on Green Spaces [51,52,53,54,55,56,57,58,59,60,61,62,63,64,65,66,67], and 12 on Active Travel to school [68,69,70,71,72,73,74,75,76,77,78,79]. No eligible intervention to reduce road or air traffic noise at school was found. Excluded studies were grouped and reasons for exclusion were recorded (Figure 2).

### 3.2. Study Characteristics

The total number of children reported in the papers ranged from 25 [63] to 57,096 [72] with a mean age of 9.1 years. The age range of children included in the evaluation was not specified in 10 studies, particularly from active travel interventions which usually evaluated aggregated data from K8 schools (5–14 years old) [42,54,64,67,68,69,70,71,74,77]. However, all these studies informed the grades that ranged from 2nd to 6th. Most recorded male/female numbers, but few reported ethnic differences [46,52,53,55,56,68,69,71].

All interventions to reduce air pollution focused on indoor school settings, classrooms, and school gyms. Green space interventions fell into two main categories: (i) increasing children’s green time: outdoor classrooms or recess time in nature by visiting parks or forests; (ii) greening schools indoors: reshaping schoolyards, implementing gardens, or installing plant walls in classrooms. Finally, by design, all active travel interventions involved permanent changes in the built environment outside the school.

In total, 16 of the 39 studies were implemented in Europe, including Denmark [50,79], Germany [49], Italy [61], The Netherlands [47,48,65,66], Portugal [67], Spain [42], Sweden [44,55], and the United Kingdom [45,60,62,63]. Most of the active travel interventions were conducted outside Europe, including the United States [68,69,70,71,74], Canada [75,77,78] and New Zeeland [72,73,76] Two interventions were held in Seoul, Korea [41,59], and one in Victoria, Australia [51].

The level of intervention implementation varied substantially across studies, from classroom level for all air pollution interventions [41,42,43,44,45,46,47,48,49,50]; at classroom [54,55,56,57,58,59,60,66,67] or school levels [51,52,53,61,62,63,64,65] for green spaces interventions. Most active travel studies involved national-wide programs evaluated at school [76,79], district [69], region [72,73], province [75], and state or multistate [68,70,71,77,78] levels. The unit of analysis was the classroom [41,42,43,44,45,46,47,48,49,50] or the child [44,45,49,50] in air pollution intervention studies, the child [51,52,54,55,58,59,60,61,62,63,64,65,66,66,67] or classroom [56,57] in green spaces studies, and individual [71,75,76,79] or aggregate level [68,69,70,72,73,77,78] in active travel studies. Sixteen studies assessed at least one health-related outcome [44,45,49,50,51,55,56,57,58,59,60,61,65,66,67,79]. Only six studies analyzed intervention effects on students’ income levels or schools from socially deprived urban areas [52,53,57,63,67,72].

Most of the included studies employed a quasi-experimental design, including uncontrolledpre-post [41,42,43,67,70,72,73,74,75,76,77,78], pre-post with cross-over design [44,45,50,54,56,57,61,62], controlled pre-post [46,47,48,58,59,60,61,67,68,71], and nonrandomized control trial with and without equivalent groups [51,55,64,65,66,69,79], and one longitudinal cluster randomized controlled trial [52], one 2 × 2 factorial group randomized controlled trial [53], one counterbalanced randomized cross-over design [63], and one cluster-randomized cross-over study [49].

Twenty-two studies evaluated the intervention less than six months after implementation, including all air pollution interventions and most green space interventions [41,42,43,44,45,46,47,48,49,50,54,56,57,58,59,60,61,62,63,64,66,67]. In total, 10 studies reported follow-up periods beyond 12 months [51,52,65,68,69,70,71,72,73,74], of which most were intervention studies on active travel.

Generally, the risk of bias was high, with no studies reaching the threshold of 9/11 for ‘high quality studies’. The median score from quality appraisal was 6 (2–8) indicating that studies were generally of weak to moderate quality (Figure 3). In addition, all studies reported insufficient information on at least one item of the quality checklist, notably concurrent exposure, representativeness, comparability, outcome assessment tools, and attrition items (Appendix A). In the following sections, we presented the specific results per intervention domain

### 3.3. Description and Impact of Interventions

A narrative synthesis approach to reporting the results was taken because of the heterogeneity of outcomes and interventions. Primary studies were grouped according to the exposure target (i.e., air pollution, green spaces, and active travel) and reported narratively, and the results are summarized in Figure 3.

The primary studies are described by intervention exposure-type according to author/brief name of the intervention, participants’ characteristics, and city/country (when mentioned), description of the intervention and comparison, outcomes, main findings, and study design label (as reported in the papers) in Appendix A. Details of the interventions using the TIDieR checklist and quality were shown in Appendix A, respectively.

### 3.4. Air Pollution

#### 3.4.1. Intervention Characteristics

All primary intervention studies (10/10 studies) targeted the reduction of air pollutants indoors by changing or regulating classroom ventilation. Two studies installed air purifiers [41,42], two manipulated the installed mechanical ventilation at schools [43,44], and six provided a new, specifically built ventilation system [45,46,47,48,49,50]. The main targeted pollutants were CO_2_ concentrations as a proxy of ventilation rates, followed by levels of fine particulates, PM_2.5_ and PM_10_. All studies provided an indirect assessment of indoor and/or outdoor air pollutants by combining different sensors to monitor pollutant levels. Therefore, information on personal levels was not available. Mostly, the experimental campaigns lasted for three [41,45,46,47,48,49,50] to eight weeks [44]. Most of the studies evaluated indoor air quality as the primary outcome, measured by the ability of the intervention to reach optimal pollutant concentrations indoors. Only four studies evaluated the impact of changes in the air pollutants indoors on children’s respiratory health [44,50] and cognitive performance [45,49,50].

#### 3.4.2. Intervention Effectiveness

The effects of improvement in indoor air quality, based on the reduction of CO_2_ levels, on pupils’ health, were inconsistent. Studies evaluated cognitive performance in terms of attention, vigilance, memory, concentration, short-term concentration and logical thinking, and accuracy by either numerical or language-based tests. A range of one [49] to nine [45] standardized paper-pencil [49,50] or computational tests were applied [45]. Two cross-over studies found significant improvement in cognitive performance after increasing the ventilation rates to a range of 6.6 to 8 L per second per person using a built-in ventilation system [45,50].

**Figure 3 ijerph-20-01746-f003:**
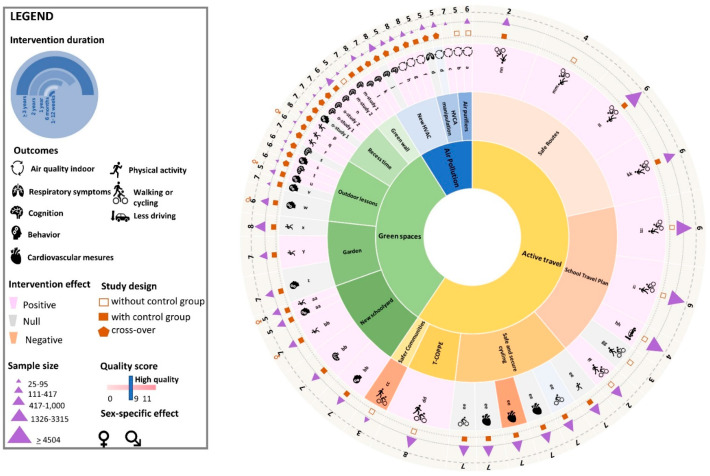
Impact of interventions per exposure target. Each slice represents an outcome. One study can have more than one eligible outcome. For more details on each intervention study see Appendix A. a = [41] effect dependent on the school location (level of nearby traffic) and year of construction (> or <15 years); b = [42] 2 school gyms from primary schools; c = [43] effect only for within late-start schools (9 AM) located near major roads; d = [44]; f = [46]; g = [47]; h = [48] not for outdoor generated pollutants (NO_2_ and PM_2.5_; i = [45]; j = [49] total number of errors in attention test increased significantly in worse condition; k = [50] significant results only when motivated students were kept, less eye pain in the recirculation condition; l = [66] only for selective attention not for processing speed; m-study 1 = [67] selective attention and working memory tests; m-study 2 = [67] only working memory tests; n = [60]; o-study 1 = [61]; o-study 2 = [61]; p = [62] playground sports increased MVPA more than nature recess; q = [63]; r = [54]; s = [56]; t = [57]; u = [58]; v = [59]; w = [55] only for general effect: positive in boys; x = [53]; y = [52]; z = [51] in contrast with qualitative data; aa = [64]; bb = [65]; cc = [76]; dd = [71]; ee = [79]; ff = [75]; gg = [78] 53 primary schools; hh = [77]; ii = [73]; jj = [72]; kk = [69] only when combined with Education component; ll = [68]; mm = [70] 79% primary K8 schools (5 to 14 y); nn = [74].

These studies reported a positive effect of the intervention on the attention and vigilance of 215 children (significative changes in 4 of 9 tests: choice reaction *p* = 0.011, colour word vigilance *p* = 0.017, picture memory *p* = 0.016, and word recognition *p* = 0.001)) [45] and short-term concentration and logical thinking (4 of 4 applied tests: addition *p* < 0.01, number comparison *p* = 0.03, grammatical reasoning *p* = 0.04, and reading and comprehension *p* < 0.01) in 79 children from different schools [50]. However, the results were only significant when “unmotivated” children were excluded from the analysis [50]. In contrast, a cluster RCT cross-over study in Germany [49] reported null effects on the short-term concentration performance of 417 students when the CO_2_ levels decreased from 2000 ppm on average to 1045 ppm, while the error rate increased by 10% in the worse condition (>2000 ppm, 1.65 increase rate (95% confidence interval 0.42–2.87).

The only study evaluating respiratory symptoms reported no differences in health symptoms or clinical signs of inflammation in the nose of 111 pupils. Children reported slightly fewer respiratory symptoms (i.e., pain in the eyes) after the increase in classroom ventilation rate, but the difference was not statistically significant [44].

All studies reported wide ranges for indoor concentrations of air pollutants, depending on the proximity of the selected schools to roads, occupancy, measurement hours, and season. The baseline concentrations of pollutants indoors were higher than recommended by national guidelines in some studies [41,45,46]. Air purifiers seemed to be effective in reducing concentrations of particulate matter (49% up to 86% reduction) and bio-aerosols (40% up to 68% reduction) in 7 childcare centers in Seoul (*p* < 0.01) [41], varying by location of the school and year of construction. Moreover, they were effective in reducing particle number and PM_1–10_ indoor/outdoor ratios (70–95%, *p*-value not reported) in two naturally ventilated school gyms, specifically when windows were kept closed and in smaller rooms [42].

Tailor-made mechanical ventilation devices, including solar-heated units implemented to control ventilation rates, seemed effective in lowering PM_10_ concentrations by two-thirds (*p*-value not reported) [46], thus lowering PM_10_, Endotoxin and ß (1,3)-glucan levels (at 1200 ppm: *p* = 0.025; 0.012; 0.002, respectively) [48], and reducing CO_2_ concentrations [47] (*p* < 0.001), but not in lowering PM_2.5_ and NO_2_ [48]. In later start schools (9 a.m.), changing the school’s HVAC system to operate one hour before rush hour traffic, reduced indoor concentrations of fine particles by 43%, and ultrafine particles by 34% (*p* < 0.05) [43].

#### 3.4.3. Specific Methodological Limitations

In general, interventions that targeted air pollution presented a modest capacity to infer causality, with poor internal and external validity, including convenience and small sample sizes, short experimental campaigns, and few health endpoints. Variations in season, window opening, HVAC system characteristics, classroom location, and comfort parameters affected the effectiveness of the interventions and few studies provided proper control for those factors [44,46,47,48]. In addition, some of them failed to achieve the planned contrasts in indoor CO_2_ concentrations between the predefined levels of ventilation [44,47,48,49].

Another common limitation was potential misclassification of the exposure due to indirect measures from a few specific sites of school (i.e., the center of the classroom) rather than personal assessment. Although 6 of 10 studies used a cross-over design [44,45,47,48,49,50] to control for individual variation, such methods did not prevent confounding due to carryover effects, transient co-exposures (i.e., breaks the outdoor, effect of noise and lighting level), time-variant confounding (i.e., season), and short-term autocorrelation within an individual [80,81]. Overall, studies provided little information on the study population (e.g., child age and sex, number of students per classroom), school characteristics such as enrolments, location (e.g., distance to roads), and the selection criteria of sites, making it difficult to judge the extent of external validity.

The quality score ranged from 5 to 8. Most studies used valid and reliable outcome tools. Notably, the most recurrent criteria missing were “5. Did the authors show that there was no evidence of a concurrent intervention which could have influenced the results?” and “6. Were the study samples shown to be representative of the study population?” (Appendix A).

### 3.5. Green Spaces

#### 3.5.1. Intervention Characteristics

In total, 10 of 17 studies focused on increasing children’s green time with outdoor classrooms in nature as the most common intervention in this category [54,55,56,57,58,59,60], followed by promoting recess time in green spaces [61,62,63]. Seven interventions were focused on greening schools indoors by implementing gardens, greening school playgrounds [64,65], and installing plant walls in classrooms [66,67] (see Appendix A).

The green space interventions varied widely across studies in amount, size, and type of green space, from installing a two-meter-high frame with eight types of green plants, [66] school gardens [51,52,53], to visiting public urban forests [60]. The duration of the interventions was usually less than one year, with the longest conducted over two years [51,52,65].

All studies justified their choice of intervention through either the Attention Restoration, Stress Reduction, Biophilia Hypothesis, or the Affordances and Loose Parts Theory [16], in which natural environments can boost psychological functioning, encourage unstructured play and physical activity, and reduce sedentary behavior. The main health outcomes studied across intervention studies were cognition and behavior problems, including working memory, classroom engagement, self-regulation, verbal conflict rates, and cooperative behaviors. Changes in Moderate to Vigorous Physical Activity (MVPA) levels were investigated in 7 of the 16 studies [52,53,54,62,63,64,65]. Outcome assessment included standardized psychological tests, questionnaires reported by children or/and teachers [57,66,67], direct observation led by researchers [52,54,57,64], and accelerometers [52,53,54,63,64].

#### 3.5.2. Intervention Effectiveness

##### Increase of Children’s Green Time during School Hours


*(a) Outdoor classrooms in nature*


Four of seven intervention studies focusing on outdoor lessons in nature had a control group [55,58,59,60]. The follow-up period ranged from one month [54] up to one year [55]. The sample size varied from 52 [59] to 230 children [55].

In general, outdoor lessons in nature took place at least once a week for an average of 95 min per session across studies, with durations ranging from 30 min [57] to six hours per lesson/day [60] (full day within a 30 min drive of the school). Outdoor classrooms were defined as adjacent areas to the school with woodland and stream [57], a portion of the schoolyard with trees, grass, sand, or mulch [58], school play-yard with trees and shade tents [56], public urban parks (i.e., National Trust sites in the UK) [60], school garden or surrounding forest and woods [55,59]. Some of the classes also included playtime in nature [59,60], specific environment and nature topics [57], inquiry-based learning activities [58], or regular math or grammar classes [56] also using the natural environment as teaching material [55].

Outdoor lessons in nature seemed to have positive effects on cognition, including educational attainment in writing (*p* = 0.002), and math (*p* = 0.047), particularly, in reading (*p* ≤ 0.001) as reported in a controlled study evaluating 223 children [60]. In addition, positive effects were reported on classroom engagement [56,57] in terms of teachers’ redirections (*p* < 0.05) or composite index (*p* < 0.01) measured by direct observation performed by teachers or researchers, as well as by standardized tests with positive results (*p* < 0.05) varying by instrument, sex (more consistent in girls), and season [58]. For peer relationships, the results were null for self-report or multiple dimensions standardized scale [55,59]. However, a decrease in behavioral problems was observed only in boys compared to girls participating in one controlled study (conduct problems *p* = 0.003, hyperactivity *p* = 0.005) [55], while improvements in self-regulation were observed only in girls in another controlled study [58]. A cross-over study found an increment in time spent in MVPA and light physical activity in 75 children compared to day lessons indoors after one month (*p* < 0.05) [54].


*(b) Recess in nature*


All three intervention studies focusing on recess in nature were cross-over designs [61,62,63] and one sub-study involved a control group [61]. The longest intervention duration was two weeks [63], with sample sizes ranging from 25 [63] up to 82 students [61].

Interventions on promoting recess time in greener areas of the school used asphalt or concrete areas as a comparison [61,62,63]. Green schoolyards were defined as areas surrounded by trees and bushes [63], a school garden [61], the school field, or green areas surrounding the school buildings [62]. Built environment comparison areas were established as hard surfaces or less green areas [61], concrete areas surrounded [63] or playgrounds with small pieces of equipment such as skipping ropes, and balls [62].

Recess time in green spaces seemed to improve the time engaged in MVPA by 40% (*p* < 0.01), notably in girls [63] compared to asphalt playgrounds. In contrast, the provision of games equipment during recess in asphalt playgrounds increased average physical activity (11.28 vs. 15.23 MVPA minutes) compared to nature-based recess (4.67 vs. 9.27 MVPA minutes) in another study (*p* < 0.001) [62]; however, recess in natural area engaged children of all fitness levels. In addition, recess time in green areas increased sustained and selective attention (*p* = 0.016) and working memory (*p* < 0.001) scores compared to the built environment condition after two weeks [61].

##### Greening Schools Indoors


*(a) School gardens*


All three intervention studies focusing on gardening at school involved robust study designs: one nonrandomized study with a follow-up from 12 up to 25 months [51], a six-month [53], and a two-year randomized [52] control trials. The interventions were implemented in 12 to 28 schools involving from 227 to 1326 children, including control groups.

School gardens were implemented in combination with a curricular component, including kitchen classes, mentoring, and garden activities such as planting, weeding, and harvesting. The gardens were defined as a 4′ × 8′ meters raised bed for each class [52], or only as a school garden [51,53] without many details.

The garden interventions showed null effects on cooperative behavior [51]. For physical activity the results were inconsistent: a 6-month RCT found no effect on self-report physical activity [53], while a 2-year RCT using accelerometers found an average increase of 58 min in moderate physical activity during garden-based lessons (*p* = 0.010), consistent with direct observation measures (i.e., PARAGON) [52].


*(b) Greening schoolyards and classrooms*


Two studies evaluated the effects of increased availability of green spaces at school [64,65] on physical activity, cognition, and behavior.

One study used a controlled before and after design to compare the replacement of four asphalt school playgrounds zones with different levels of green, including introduction of trees, mulch, and boulders in two zones, only grass and trees and outdoor decomposed granite floor, mulch, and plant border in the fourth zone [64]. Greening playgrounds seemed to improve MVPA participation (41 to 55% of students observed, *p* = 0.003) in both accelerometers and direct observation measures, which remained higher in girls after a 4-month follow-up (*p* < 0.05). Furthermore, a decrease in verbal conflicts was observed after the same period in both sexes (*p* < 0.001) [64].

The second study performed a longitudinal intervention comparing the replacement of some paved areas with grassy hills, bushes, trees, and tunnels made of tree branches with paved schoolyards [65]. After two years of follow-up, 351 students (n = 355 control at baseline) from the intervention school improved their attention test scores (*p* < 0.05) and presented fewer peer problems (*p* < 0.05). Furthermore, accelerometer measures showed increased physical activity only for girls (by 20% to 35% MVPA during recess, *p* < 0.001) [65].

Regarding the visibility of green spaces, two studies evaluated the effects of the installation of a green wall in a classroom on attention and working memory. Both evaluated the intervention effects after two months and included control groups (i.e., control classroom), except one sub-study performed with a before-and-after design [67].

In one study, a green wall was defined as 1.25 m wide and 2 m metal frame with layers of felt, which provide fertile soil for the plants with 8 types of green plants, including Spathiphyllum, Philodendron, and Dracaena in one classroom. An increase in selective attention was observed (*p* = 0.035) although processing speed was not changed after 2 months in a sample of 170 children [66].

In the second study, first the green wall was composed of artificial green and one month after vegetable pots were introduced as part of the intervention [67]. Sustained and selective attention (*p* < 0.001) and working memory (*p* = 0.012) significant increase in the intervention group (n = 55) compared to the control classrooms (n = 40), with a positive cumulative effect after the introduction of vegetable pots (two months after). The impact was similar for working memory in students from low- and middle-income schools in a sub study involving two primary schools (total sample n = 75, *p* < 0.001) [67].

#### 3.5.3. Specific Methodological Limitations

Intervention studies on green spaces showed weak to moderate quality. Most studies included a control, but evaluated only short-term changes in cognition, behavior, and physical activity. All studies failed to provide information on the quality of green spaces, most roughly describing the type of vegetation, and none used a specific tool to distinguish between degrees of “greenness” (i.e., Normalized Difference Vegetation Index, inventory). Furthermore, most studies failed to provide evidence of the absence of a concurrent intervention which could have influenced the results (Appendix A), except for [51,53]. The differences in exposure levels between control and intervention groups were not clear in some studies [58,61,62,63]. In addition, the levels of greenery reached after reshaping interventions varied considerably and some studies reported pooled results without considering those differences [64,65]. For behavioral problems, a range of measurements was used, notably for classroom engagement and off-task behavior including direct observation, self-report questionnaires by parents, students, or teachers, and some standardized scales (Appendix A).

Overall, the quality score ranged from 5 to 8, with 10 of 17 studies scoring ≥ 7. The three checklist criteria which were the most recurrently missing were: “5. Did the authors show that there was no evidence of a concurrent intervention which could have influenced the results?”, “6. Were the study samples shown to be representative of the study population?”, and “8. Were numbers of participants at follow-up identifiable as at least 80% of the baseline?” (Appendix A).

### 3.6. Active Travel

#### 3.6.1. Interventions Characteristics

Intervention studies to promote active travel to school involved so-called Safe Routes to School [68,69,70,74], School Travel Plan [72,73,75,77,78], and other local programs based on safety measures for active school commutes [76,79]. In general, these programs are based on the built environment and individual behavior interventions, commonly defined as the 4E’s components: encouragement, enforcement, education, and engineering. Engineering components included installation or improvements of infrastructures surrounding schools such as sidewalks constructions or reparation [68,69,71,74,75], path upgrades [76], speed bumps, signage [72,73,74,79], installation of bike racks or bicycles lanes [70,77,78], and yellow school zones [75].

Essentially, active travel was defined as walking (i.e., walking partway or walking buses) or cycling to/from school in all studies, non-motorized scooting [72,73,75,76], and less driving [77]. One study focused exclusively on cycling [79].

The main outcome was the proportion of students walking or cycling to school or changes from passive to active travel modes. Hands-up classroom surveys led by teachers were the main instrument to assess the outcome. Some studies have analyzed secondary data from program coordinators or funding institutions’ reports [68,70,74]. Other instruments included questionnaires to parents [68,71,72,73,75,78] and direct observation of the number of bikes and pedestrian traffic around the schools, including from video cameras [76]. Only one study from Denmark evaluated the effect of the intervention on health using self-reported overall biking, overall physical activity, obesity, and cardiorespiratory fitness [79].

In The studies differed in terms of follow-up length from one year [75,76,77,78,79], three to four years [69,71,72,73], and up to 5- and 10-years retrospective panel evaluations [68,70,74]. The sample size across studies varied widely. One-year evaluation studies ranged from 123 [76] up to 2401 [79] children, with one retrospective study evaluating more than 7000 unique parent questionnaires on their child’s active travel [78]. Long-term evaluations mainly included large-scale interventions with sample sizes ranging from 1999 [74] up to 57,096 [72].

#### 3.6.2. Intervention Effectiveness

The effects of interventions to improve active travel to school were inconsistent depending on the study duration. Long-term controlled study found an absolute increase from 5 to 20% in walking and biking after 4 years when combining encouragement and infrastructure actions (*p* < 0.05) [69]. A three-year controlled study found modest effects on active travel in the morning comparing schools with and without infrastructure intervention [71]. In addition, a pre-post uncontrolled study found a very low increase in active travel from 41% to 42% (OR = 2.65, 95% CI = 1.75–4.02) after 3 years of implementation, independent of the school size, but greater in students from high socioeconomic status (38.9% to 39.1% increase by the third year, *p* = 0.0078) [72].

At the state level, one retrospective study found an absolute increase of three percentage points in active travel associated with engineering improvements in intervention schools compared to control schools after five years (*p* = 0.031) [68]. Similarly, another 5-year study observed significant increases in active travel across all the 4 states investigated (48 completed projects, 53 schools affected by the completed projects), with an overall increase in the proportion of active travel from 12% to 17% (*p* < 0.001) (walking from 8.8% to 13.3%, and bicycling from 2.0% to 3.2% [70]). A 10-year mode shift study found that living within 250 ft of the Safe Routes to School project increased the probability that a child walked to school [74] (*p*-value not reported for this result).

For short-term evaluations of active travel interventions, the picture was less consistent. For instance, one year before and after studies have shown contrasting results. One found positive effects on active travel rates, but with inconsistencies between students (43% vs. 45%, before vs. after) and parents’ reports (37% vs. 43%, *p*-value not reported) [75]. Another reported null effect with a proportional decrease in walking to school [76], as well as a national-level study [78], although post-intervention rates of active travel varied across schools. In contrast, a post-intervention retrospective study observed a decrease of 17% in car use in the same study population (*p*-value not reported for this result) [77]. Older children, those living less than 500 m from the school, and those from middle class neighborhoods compared to low SES schools, were significantly more likely to change from car transport to active travel (*p* < 0.001).

The only study evaluating the impact of the active travel intervention on health found a change in cardiorespiratory fitness in an unfavorable direction in the intervention group compared to the control, probably due to implementation delays and an unbalance between controls and intervention groups related to the outcomes. No statistically significant differences were found in leisure-time physical activity, school cycling, or risk of overweight/obesity [79].

Some controlled studies provided information about which components of the intervention were indicative of active travel changes. Receiving only education and encouragement components was associated with a non-significant increase in walking and a five-percentage point increase in biking after four years. From the same study, improving sidewalks and crosswalks had a non-significant impact on walking and biking, although the study failed to measure this specific-component impact due to implementation delays. In addition, the same study reported that the combination of augmenting education programs with built environmental improvements was associated with increases in walking and biking of 5–20 percentage points [69].

Another study found few differences between providing larger amounts of funding for infrastructure projects compared with smaller amounts. However, non-infrastructure funding appeared to have slightly negative effects on active travel over time compared to a matched control [71]. At the state level, education and encouragement programs were found to be cumulative, with each additional year of program participation associated with an absolute increase of 1% in the proportion of students walking and bicycling per year, while engineering improvements lead to a 3% increase after 5 years [68].

In summary, combining education programs with engineering improvements seemed to be the best approach considering that built facilities may support behavioral change over time. Shorter residential distance to the intervention area [74,76], older children and grades [69,73], smaller schools and higher socioeconomic status [73], safety perceptions in parents and children [76], and long-term interventions seemed to be relevant predictors of success in the included studies.

#### 3.6.3. Specific Methodological Limitations

The inconsistencies in results between studies may be due to differences in the scale of evaluation from the school level to the national level, in study duration ranging from 1 to 10 years, in the use of aggregate data instead of individual-level measures, in the level of intervention implementation across schools, as well as in the type of infrastructure improvements. The aggregated effect may dilute school characteristics and local features relevant to the success of the programs, which prevents conclusions on which aspects of the intervention work for which context.

Information on fidelity or adherence (i.e., the extent to which an intervention was delivered as conceived and planned) was lacking in most studies, except in five interventions on active travel [69,71,72,73,77,78,79] where brief information was provided (Appendix A), including valuable insights from qualitative data [71]. Many schools may not follow the same standard planning process as informed in the intervention program [72,73], and because of unforeseen issues such as political, budget, or administrative changes [77,78,79] they may implement the intervention at different times [69,71]. Those issues may affect the study’s ability to attribute changes to the intervention. In controlled studies, more information about concurrent interventions is needed due to the difficulty to guarantee the control group was not offered any intervention or was affected indirectly by the intervention under study. Built environment changes may have a buffering impact affecting other schools and children. Furthermore, residual exposure in control groups is not clearly described: nearby schools and respective students can be affected by structural improvements (i.e., buffering effect) [74]. In the same way, it should be considered that students from the same intervention school can have different levels of exposure depending on where they live, the route they take to school, and sociodemographic characteristics.

Few studies controlled for neighborhood characteristics such as walkability score, season or weather [68,71], and distance to school [69,74]. All studies used self-report measures as the main outcome and discrepancies between students’ and parents’ self-report measures have been observed [75].

Overall, the quality score ranged from 2 to 8, with 6 of 12 studies scoring ≥ 6. In addition to missing information on concurrent interventions and representativeness, the most recurrent criteria missing were “8. Were numbers of participants at follow-up identifiable as at least 80% of the baseline?”, and. “9. Were valid and reliable tools used to assess participant outcomes?” (Appendix A).

## 4. Discussion

To our knowledge, this systematic review is the first of its kind to provide a broad appraisal of the literature on school-based interventions evaluating the effectiveness of different built and natural environment modifications to improve child health, health behaviors, or exposure levels. We identified three main intervention types: interventions to improve indoor air quality by improving ventilation rates, interventions to increase the number of time children spent in green space, and interventions to increase active travel to school. We found no studies exploring interventions to reduce noise stressors, despite this being an important factor related to poor health among children.

The 39 studies included in this systematic review demonstrate that there is modest evidence to support the effectiveness of these school-based interventions on children’s health, behavior, or exposure levels. In summary, we found: (i) reductions in levels of indoor generated air pollutants in classrooms through built-in ventilation devices, air purifiers, and changes in the timing of building ventilation system; (ii) inconsistent effects of higher indoor ventilation rates on cognitive performance; (iii) inconsistent effects of more time spent in green spaces or greening schools indoors on behavior, but positive effects on cognition dimensions, particularly attention, and consistent increases in MVPA, especially in girls; and (iv) a relatively small positive effect of multi-component active travel interventions on biking and walking after long-term (≥3 year) interventions, and inconsistent and even negative effects on cardiometabolic health and physical activity from 1-year interventions.

### 4.1. Risk of Bias and Study Quality

Overall, the studies presented poor internal validity due to study design, data collection by use of non-reliable and valid instruments, convenience sample and poor description of the recruitment process of schools and students, and weak reporting concerning the intervention features (i.e., staff, training, materials used, modifications, and fidelity), and characteristics of the population (children and schools) (Appendix A).

Furthermore, we found that interventions were often poorly described (Appendix A). Few studies [49,52,53,69,71,72,73,77,78,79] commented on adherence or fidelity to know if the intervention was delivered as planned. Only two measured and provided the fidelity of the intervention implementation, based on self-report instruments, but without considering it in the analyses [53,79]. Researchers should report information on intervention adherence and fidelity to avoid error type III (a potential null impact of the interventions not caused by implementation failure) [82] and when possible, evaluate the impact of each component of the intervention (what works for whom, how, and under what circumstances). In some cases, as in quasi-experimental studies, the level of implementation depends on policy funding, schools’ calendar, and other unforeseen situations. However, the collection and reporting of information on the level of implementation are crucial to compare groups and establish the intervention impact. This information should be systematically measured even retrospectively and always discussed, including when it was not measured, in that case at least as a limitation of the study.

Details on the materials, costs, procedures and training of staff involved [41,42,44,45,46,47,48,49,50,61,62,63,64,65,66,67,71,76] as well as on modifications of the intervention components, were missing for a great part of the studies, except for [51,53,57,58,61,79]—these aspects are essential to inform future interventions and enhance comparability. Furthermore, there was little reporting on whether teacher and student preferences were measured and considered (Appendix A).

Concurrent interventions were not considered or insufficiently described in the majority of the studies, except for some studies which discussed it at some level [43,45,47,48,50,51,53,68,71,72,76,79] (Appendix A). For instance, secular trends [71], personal exposures out of school, and unforeseen interventions in the control groups could represent major confounders. Studies should provide exhaustive information on how those factors were considered and minimized, or at least describe the impossibility to measure them and discuss clearly how these limitations may affect their results. For all exposures under the scope of this review, seasonality is an important confounder– air pollution peaks, greenness, and prevalence of active travel may be influenced by weather, temperature, and other seasonal conditions and few studies considered those aspects [58,69,71].

The majority of included studies are prone to selection bias due to intervention placement self-selection–giving resources only to schools willing to participate–and attrition bias due to different rates of loss to follow-up [44,45,47,48,51,52,53,55,63,64,65,75,77,78]. Indeed, many studies used convenience samples (Appendix A). Few studies provided comparisons between participants and non-participants, and inclusion and exclusion criteria were mostly omitted [41,42,44,45,46,47,50,56,57,60,61,63,69,70,72,74,77,78]. More information on the population and school characteristics, as well as on dropouts, is needed to inform on external validity and the extension of attrition bias. For studies using different panel data (i.e., different children taking part in the data collection period) it is important to explicitly explain how those groups overlap—and if not, why that occurred.

Overall, studies included reliable and validated outcomes tools to measure health outcomes and physical activity, notably less frequently for active travel interventions and behavior (i.e., self-regulation) [55,58,65]. Particularly for active travel studies, hands-up surveys were a common way to measure changes in travel mode based on the teacher’s count, although the level of uncertainty related to the source (i.e., secondary data in intervention reports), double-counting, and students’ absenteeism need to be considered (Appendix A). We reinforce the recommendation of objective measures in this field raised by previous reviews [83].

Cognitive development was the most common health outcome as well as physical activity, with only two studies evaluating respiratory symptoms [44,50] or cardiometabolic outcomes [79]. Mostly, the time between the intervention and effect measurement was short, especially for air pollution and green space interventions (Appendix A). It is plausible that changes in the concentration of air pollutants and psychological restoration effects of green spaces can occur after only a short exposure [84]. However, changes in health endpoints and behavior over the long term are still a gap and, in some cases, the time between implementation and evaluation seemed insufficient to rule out novelty and learning effects. Particularly, physical activity and active travel related to environmental interventions may have a lag period of change.

In general, robust controlled studies, high-quality longitudinal studies, and randomized controlled trials are lacking. Natural experiments are methodological alternatives to randomization providing robust evaluation conditions [35]. However, real-world interventions related to the urban environment are hard to adhere to random allocation principles. Usually, researchers are dependent on policy and local authorities’ agendas which included unforeseen changes especially when involving structural improvements. Robust analysis methods, such as propensity scores matching [85] to mitigate the heterogeneity between the comparison groups in terms of at least observable confounders (i.e., sociodemographic, socioeconomic, and school characteristics) are recommended. Future studies should consider including longer follow-ups, standardized tools to assess health-related outcomes, behaviors, and exposure levels, and qualitative approaches to identify the mechanisms shaping successful interventions.

### 4.2. Future Directions

The interventions we reviewed related to air pollution gave some evidence for reductions in levels of indoor air pollutants in classrooms following ventilation and purification interventions, but evidence for effects on health is scarce and we found inconsistent effects of higher indoor ventilation rates on cognition performance. Therefore, future air pollution school intervention studies should focus on including health-related endpoints (i.e., respiratory and neurodevelopment). The studies we reviewed relied mostly on indoor monitoring for air pollution exposure assessment and CO_2_ as a proxy of air quality; exposure assessment can be improved through the inclusion of personal monitoring approaches and key pollutants should be monitored (i.e., traffic-related pollutants), not only ventilation rates. It is important also to consider the assessment of pollutant mixtures and their variation across space and time. Beyond describing only building and classroom characteristics, future studies should better describe population features, including at least child sex, age, grades, and the number of students in the monitored classroom. This can enhance the ability to compare and replicate interventions. More information on residual exposures related to time spent in other areas of the school when the intervention is not operating is needed. Interventions implementing new ventilation systems should report the related costs, maintenance, and feasibility aspects (Figure 4).

Evidence from green spaces intervention studies is promising in terms of health benefits and increased physical activity, especially in girls. However, for behavioral problems from the studies included in our review, the evidence is inconsistent. Although mainly studies included control groups, the sample sizes were small with short follow-up periods and poor descriptions of the type, quality, and amount of vegetation modified, hindering replicability. Future studies should assess and report the quality, amount, and type of green spaces in schools and account for residual confounding, as well as the dose-response thresholds. Other sources of nature at the school ground or surroundings should be assessed and reported especially in control groups. The sex-specific effects of outdoor lessons and recess in nature on behavior and cognition need to be clarified. Studies evaluating the impact of green interventions on behaviors such as self-regulation, peer relationships, classroom engagement, and attention should prioritize standardized instruments. Longer follow-ups and increased sample sizes are recommended to minimize Hawthorne effects and clarify the long-standing benefits of green space. Furthermore, it is still unclear which specific components may promote more nature-related health outcomes and behaviors (Figure 4).

The intervention studies included in our review gave some evidence that the rates of walking and cycling commute can be enhanced by interventions involving changes in the built environment around schools. However, this effect seemed to be small and more consistent in interventions longer than one year. Rates of active travel were usually self-reported from different sources showing inconsistencies and prone to response-shift bias, therefore, objective measures are required. Intervention studies in this domain should measure and report the details of the infrastructure and programs implemented before the main intervention—within and surrounding schools. Few studies informed on the facilities (i.e., walkability) or aspects of safety, air pollution, and traffic flow in the school environment, nor on a walkable residential distance to school. There is little robust evidence on the health benefits of active travel in children. Studies should also consider measuring the impact in terms of levels of physical activity and included related health endpoints (i.e., cardiometabolic). Infrastructure is costly and permanent changes in the landscape should be specifically evaluated at individual and school levels (Figure 4).

Finally, no intervention studies were identified on road-traffic noise. Vehicle and aircraft noise is recognized as a health threat, especially increasing the risk of cognitive impairment in children, disturbing the classroom environment, communication, listing, and reading tasks [86]. Studies should target interventions or at least the assessment of road-traffic noise, which can be explored from the angle of air pollution and green space interventions, investigating neurodevelopment outcomes, their mechanisms, and pathways.

Future studies would benefit from considering the impact of built and nature interventions on multiple exposure levels—urban exposures are not static and may affect each other. Most intervention studies focused on only one exposure-outcome effect ignoring potential adverse effects of other exposures or potential cross-benefits. For instance, it is not clear how specific travel modes, including walking and cycling, increase children’s exposure to air pollutants, green spaces, and noise on their way to school. What is the health impact of walking or cycling to school on a green route or close to a busy road? Does active travel or outdoor classes especially during rush hours increase personal exposure to air pollution and noise? Although physical activity benefits can mitigate the deleterious effects of increased inhalation of pollutants, this dose seems to be higher among cyclists and pedestrians than among commuters using motorized transport [87], and this is particularly important for children. On the other hand, increases in active travel to school may also impact car dependency, reducing air pollution, and noise around schools—but how big is that impact? Furthermore, increasing green space around schools may reduce pollution—to what extent? Future interventions would benefit from exploring how these elements affect each other to potentialize intervention impact in promoting a healthy school environment.

Finally, few of the reviewed studies considered effects in specific population subgroups, such as socioeconomically deprived groups. It is already well known that children from deprived neighborhoods are more exposed to urban hazards, and may be more prone to commute, play, and study under adverse conditions. In this review, we identified only six studies that somehow reported interventions targeted at low-income schools’ subgroups or across income levels [52,53,57,62,67,72]. Future interventions should consider these aspects and measure the impact by reporting at least the deprivation level in a school zone, the social-economic status of the enrolments, and how the intervention is intended to mitigate inequities (or not).

### 4.3. Policy and Practice Recommendations

Despite the limitations of the literature reviewed, we can provide some recommendations for policy and practice (Figure 4). There is sufficient scientific evidence of the burden of air pollution, noise, lack of green spaces, and physical inactivity on health, especially in urban settings. Interventions to mitigate those urban hazards are essential, especially during vulnerable life stages such as childhood. Intervention developers should work with teachers, students, and parents in a co-productive way to ensure that interventions and approaches are acceptable and feasible [88].

Improving indoor air quality may lead to improvements in health [3]. The studies included in our review indicated that relying on air purifiers in schools close to busy roads and increasing ventilation rates at schools with HVAC systems seems to be a short-term alternative to improve air quality indoor classrooms. However, promoting structural changes by retrofitting ventilation in the school building seems difficult and limited and the cost-benefit is not clear. For new schools, planning control policies can be an effective way of ensuring that schools are cited in low traffic, low pollution neighborhoods. Future interventions should also consider reducing pollution around schools and improving indoor and outdoor air quality, for example, through clean air zones or green barriers [4].

Policymakers should explore ways to maximize the time children spend in outdoor, natural settings, with a minimum time of 30 min per week. This could be promoted by the greening of school playgrounds, or by the use of ‘forest schools’ or similar where children regularly experience these natural environments.

For active travel, it seems that multicomponent interventions are more efficient, and built environment changes (for example infrastructure and support for bikes and walking) are necessary to support behavioral changes. In addition, for these interventions, school-level characteristics are key in designing the intervention, considering students’ grades, local culture, and needs. Evaluation is essential and monitoring and evaluation should be part of the policy cycle.

### 4.4. Strengths and Limitations

This systematic review included a comprehensive search across school-based interventions on the built and natural environment to promote and protect children’s health. Results were limited to papers published within the last 11 years (2010–2022) to ensure records were relevant to the present day, and to those that had taken place in Europe and high-income to emulate comparability in terms of urbanization levels and education structure. We used the TIDieR framework as a concise and comprehensive reporting structure.

The broad review question led to high heterogeneity inherent to environmental studies which restricted our ability to draw robust conclusions. However, we provided a comprehensive description of intervention features and gaps to inform and guide future studies. Future reviews should tackle grey literature, data from local councils on unpublished intervention evaluations, and behavioral interventions with potential impacts on traffic load, road traffic air pollution, and noise.

We did not restrict studies based on the length of the follow-up reflected in the inclusion of the high number of short-term studies. The lack of longitudinal intervention studies limited our ability to inform on the sustainability and effectiveness of interventions over time. However, short-term effects of the reduction in air pollution, and an increase in exposure to green spaces were demonstrated and are valuable for children’s health.

Furthermore, we did not exclude studies based on quality and therefore conclusions should be interpreted with caution. Finally, due to the heterogeneous nature of the review, outcome measures, and interventions, it was not appropriate to conduct a meta-analysis.

## 5. Conclusions

This systematic review shows modest evidence that school-based built and natural environment interventions can improve children’s health, physical activity, active travel, and exposure levels to air pollution and green spaces. Specifically, there is some evidence that increasing ventilation rates improve indoor air quality in classrooms, that green space interventions have potential health benefits, and that safe route programs with infrastructure improvements lead to an increase in active travel. Future research should focus on incorporating objective exposure and outcome measures, control groups, and health endpoints in long-term evaluations.

## Figures and Tables

**Figure 2 ijerph-20-01746-f002:**
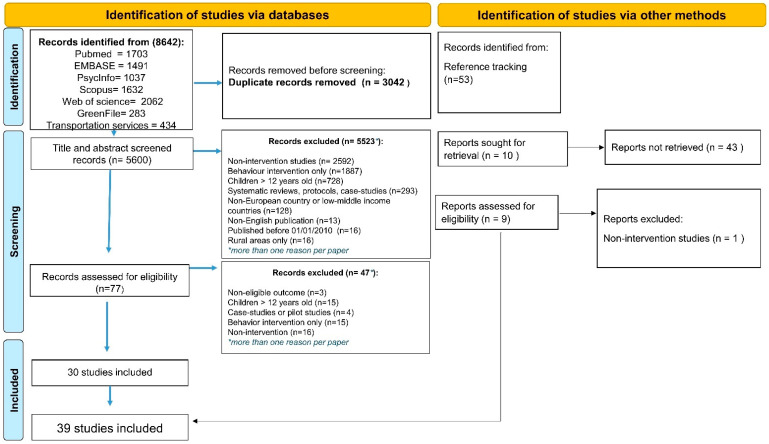
PRISMA flow diagram of study search and selection [40].

**Figure 4 ijerph-20-01746-f004:**
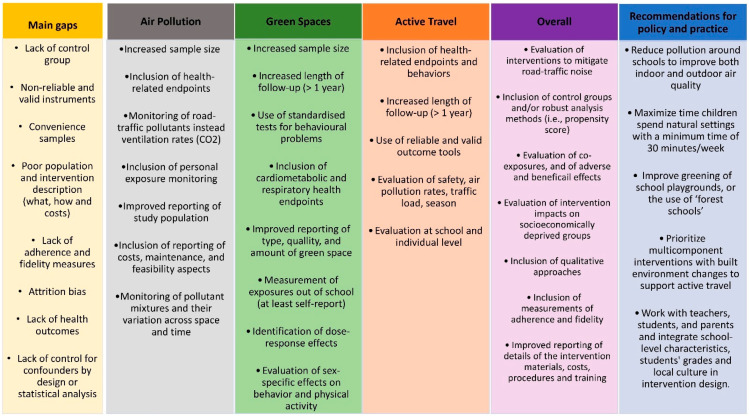
Recommendations for future studies and policymakers from each intervention target.

## Data Availability

No new data were created or analyzed in this study. Data sharing is not applicable to this article.

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
