# Peer review of "School-Based Interventions to Support Healthy Indoor and Outdoor Environments for Children: A Systematic Review"

_ijerph, 2023, doi:10.3390/ijerph20031746_

Round 1

Reviewer 1 Report

The authors critically review data from a number of published reviews which examined the effectiveness of different types of intervention to improve the school based environment for young children.   The various studies although their design was weak did show that the various interventions employed did result in a small improvement.  The review will valuable aid to further work in this field but can improved if the following changes are considered.

1.  All 4 figures need to be in a larger font so that they are readable and probably be presented in the landscape orientation. 

2. The authors are critical of the study designs that have been published so far and do mention the  various defects however it would be very useful to the reader if they could summarise these weakness and how they can be avoided in future.

Author Response

Reply to Reviewer 1

Reviewer 2 Report

It is my greatest honor to review this manuscript of a systematic review on school-level intervention to improve indoor and outdoor environments for children. The authors investigated school-level interventions for children aged 5-12 in European and high-income countries. They included thirty-nine papers with three main intervention types and provided critics and summarized future directions. I have some suggestions for the authors, please see them listed point-by-point below.

Introduction

1. It would be helpful if authors could provide evidence on current knowledge on how the built environment influences the health outcomes they listed in the review.

2. The authors could provide some examples of "school-level interventions" to improve indoor and outdoor environments and their influences on children's health. It would be particularly helpful if they could provide some results in the current literature to strengthen the need for a systematic review.

Methods

1. The authors included multiple outcomes, including children's physical and cognitive health, exposure level changes, and behaviors related to physical activity and active travel. I would recommend the authors narrow down the outcomes a little bit to focus on children's physical and cognitive health. Also, the exposures that interventions worked on varied greatly, from air quality to green space, to built environment. It is quite difficult to understand the main research question of this study.

2. Also, active travel is a concept that may be novel to some readers, therefore, the authors could provide some information in this part.

3. Did the authors consult a librarian for the search terms and search strategy?

4. The authors only included peer-review papers published in English from 2010 onwards; why limited the studies after 2010?

5. Why the authors only included studies with children aged 5-12? Is there any rationale for this decision?

Study Summaries

1. It would be helpful to provide statistical data, for example, whether the comparison is statistically significant. 

2. It would be helpful to the readers if the authors could provide references when discussing and providing critiques for the studies.

3. The levels of intervention of the included studies varied; it would be helpful if the authors could provide information on the level of intervention in the summary.

Author Response

Thank you. Please find the point-by-point answer attached. 

Reviewer 3 Report

Can be improved the methodology description 

The result can be more specific

Its possible to make more conclusions 

Author Response

Please find the point-by-point answer attached. 
